# Mediation role of low birth weight on the factors associated with newborn mortality and the moderation role of institutional delivery in the association of low birth weight with newborn mortality in a resource-poor setting

Rornald Muhumuza Kananura  [1,2,3]

[1]Department of International Development, London School of Economics and Political Science, London, UK
[2]Department of Health Policy Planning and Management, Makerere University School of Public Health, Kampala, Uganda
[3]Center of Excellence for Maternal and Newborn Health, Makerere University School of Public Health, Kampala, Uganda

**Correspondence to**
Rornald Muhumuza Kananura;
r.m.kananura@lse.ac.uk

## ABSTRACT

**Objectives** To assess low birth weight's (LBW) mediation role on the factors associated with newborn mortality (NM), including stillbirth and the role of institutional delivery in the association between LBW and NM.

**Design and participants** I used the 2011–2015 event histories health demographic data collected by Iganga-Mayuge Health Demographic and Surveillance Site (HDSS). The dataset consisted of 10758 registered women whose birth occurred at least 22 weeks of the gestation period and records of newborns' living status 28 days after delivery.

**Setting** The Iganga-Mayuge HDSS is in Eastern Uganda, which routinely collects health and demographic data from a registered population of at least 100000 people.

**Outcome measure** The study's key outcomes or endogenous factors were perinatal mortality (PM), late NM and LBW (mediating factor).

**Results** The factors that were directly associated with PM were LBW (OR=2.55, 95% CI 1.15 to 5.67)), maternal age of 30+ years (OR=1.68, 95% CI 1.21 to 2.33), rural residence (OR=1.38, 95% CI 1.02 to 1.85), mothers with previous experience of NM (OR=3.95, 95% CI 2.86 to 5.46) and mothers with no education level (OR=1.63, 95% CI 1.21 to 2.18). Multiple births and mother's prior experience of NM were positively associated with NM at a later age. Institutional delivery had a modest inverse role in the association of LBW with PM. LBW mediated the association of PM with residence status, mothers' previous NM experience, multiple births, adolescent mothers and mothers' marital status. Of the total effect attributable to each of these factors, LBW mediated +47%, +15%, +100%, +54% and −45% of rural resident mothers, mothers with previous experience of newborn or pregnancy loss, multiple births, adolescent mothers and mothers with partners, respectively.

**Conclusion** LBW mediated multiple factors in the NM pathways, and the effect of institutional delivery in reducing mortality among LBW newborns was insignificant. The findings demonstrate the need for a holistic life course approach that gears the health systems to tackle NM.

## STRENGTHS AND LIMITATIONS OF THIS STUDY

⇒ The study used a large event history health demographic data that had never been exploited to study perinatal and newborn health.
⇒ The gestation age data that could indicate if a delivery was premature or otherwise and if a pregnancy loss/death was a stillbirth or otherwise were not collected during birth registration.
⇒ There was no information on the maternal morbidities and complications that lead to different birth outcomes, which could have been used to understand their effect along the path.
⇒ There was a considerable proportion of missingness for some of the variables addressed through multiple imputations to minimise the missingness bias.

## INTRODUCTION

The evidence is clear that the availability of maternal and newborn life-saving technologies and effective interventions have worked towards reducing newborn mortality in developed countries and other developing countries.[1–3] However, newborn mortality continues to have the largest portion of under-five mortality (47%) and has remained unacceptably high in sub-Saharan Africa (SSA).[4–7] Of the 2.5 and 2.4 million newborn deaths in 2018 and 2019, respectively, 42% was a portion of SSA in both years of reporting.[4 5] The neonatal mortality rate in SSA has persistently remained unacceptably high at an average of 27–28 per 1000 live births.[4–7] At least 75% of the newborn mortality occurs within the first week of life,[8] and close to 55% within 24 hours.[8] These rates could be underestimated as stillbirths are normally misclassified because of weak documentation systems in SSA.[9]

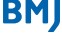

The high proportion of newborn mortality has been attributed to low birth weight (LBW)[10 11] and their related consequences.[12] The direct effect of the socioeconomic and demographic factors such as wealth, education level and maternal age on newborn mortality indicated in existing frameworks[13 14] has been widely studied; however, evidence of how these could be mediated by LBW is elusive. For instance, a direct effect of adolescent age and advanced maternal age on neonatal mortality has been observed in some studies.[13–16] However, maternal age extremes are associated with the increased risk of LBW and other adverse complications,[17–23] ultimately leading to newborn mortality. Regarding socioeconomic factors, educated and less poor women are more likely to access better maternal health prevention services that control LBW[24–27] and can afford to incur an extra bill, aside from what is freely provided by the government.[28 29] Other LBW contributors include morbidity in pregnancy (syphilis and malaria), prepregnancy and pregnancy malnutrition exposure and exposure to environmental factors, particularly indoor air pollution,[10 19 30] which are common among vulnerable populations. The scarcity of evidence on how LBW mediates newborn mortality limits the design of moderating interventions. Understanding the mediating role of LBW in the association of newborn mortality with socioeconomic, demographic and other individual factors may contribute to generating clinical and community-based interventions that focus on women's health before and after pregnancy, and after delivery.

Furthermore, access to health services is the most effective intervention in preventing maternal morbidities and controlling LBW and other complications' consequences.[1 2] Attending the recommended number of antenatal care exposes the mother to the interventions for identifying and managing pregnancies[31] that may lead to LBW. Additionally, access to ANC exposes the mother–fetus dyad to preventions and care interventions that reduce maternal morbidities associated with the increased likelihood of LBW and prematurity. The first day of life is vital for the survival of newborn, and thus, access to quality of care within labour or delivery time is crucial. During childbirth, women should be attended to by skilled health workers, who should assess the labour complications, including obstructed labour conditions that may need emergency services such as caesarean section.[32–35] After delivery, the newborns should be screened for life-threatening signs, including LBW and asphyxia, to benefit from health interventions such as resuscitation for asphyxia and corticosteroid treatment administration/kangaroo mother care for LBW.[1 2 34 36] Notably, reaching the health facilities in the SSA context does not guarantee access to the required services[24 37] because of multiple (concurrent) barriers that impede access to needed services. The health facility-based interventions could be performed inappropriately or too late, or women and newborns may fail to access the necessary interventions at all because of inadequacies in supplies and skilled health workers.[3 38] With the high increasing rates of LBW and preterm birth vis-à-vis the stagnation of high neonatal mortality rates in SSA despite the increase in health facility delivery, evidence on how the health facility deliveries moderates the survival of LBW newborns is warranted.

The previous discussion highlights the role of LBW in mediating the association of some of the factors with newborn mortality and the role of institutional delivery in the relationship between LBW and newborn mortality. However, the pathways and mechanisms of the interactions are unclear, particularly in the sub-Sharan African context. In this study, I analyse the Iganga-Mayuge Health and Demographic Surveillance Site (HDSS) data to assess the LBW's mediation role on the factors associated with newborn mortality and the role of institutional delivery in the association between LBW and newborn mortality. These data have not been exploited in studying newborn survival in Uganda and other countries that share the same context.

## METHODS
### Data
In this study, I used the 2011–2015 event histories (EH) health demographic data collected by Iganga-Mayuge HDSS. The EH dataset consisted of 10 758 registered women whose birth occurred at least 22 weeks of the gestation period and records of newborns' living status 28 days after delivery. The HDSS data are collected in the eastern region of Uganda, covering at least 185 villages in seven subcounties of Iganga and Mayuge districts. The HDSS routinely collects data on pregnancies, births, migrations and mortality from a registered population of at least 100 000 people. Community health workers who are based at a village level notify and report all events, and thereafter, the HDSS Field Assistants follow up all the reported events for actual documentation using standard tools. The analysis was limited to births among residents, in-migrants who stayed in the HDSS for at least 6 months and outmigration that had stayed outside the HDSS within 6 months. The definition of an HDSS resident is someone who has stayed within the HDSS for at least 6 months, and outmigrants remain the HDSS residents within 6 months of outmigration. To date, such large event history data in a resource-poor setting have not been used for this kind of studies.

### Study variables
The study's key outcomes or endogenous factors that were used to describe the perinatal and newborn mortality or survival system are perinatal mortality, late neonatal mortality and LBW (mediating factor). Perinatal mortality includes deaths within 7 days after birth and all stillbirths, while late neonatal mortality is defined as death within 21 days after the first week of survival (7–28 days),[39 40] describing the timing of death within the newborn survival system. LBW, defined as a birth weight

that is less than 2.5 kg at birth,[41] explains how newborn survival could be affected indirectly by other factors through birth weight.

The newborn adverse outcomes' exogenous variables were categorised as maternal and household factors (having a partner, maternal education level, maternal age and household wealth index), child individual risk factors (childbirth order, birth category – multiple or singleton and child's sex), institutional delivery, mother's previous experience of newborn loss or stillbirth, place of residence (rural or urban) and birth season (in annual quarters). Having a partner was categorised as a dummy variable for 1: having a partner (those who were either married or cohabiting) and 0: having no partner (those who were single or widowed). The wealth indices 3–4 were considered less poor, and wealth indices 1–2 were considered poorer. The wealth index was generated using principal component analysis, and the items were household assets, household roof structure, household floor structure and household wall structure. Birth order was categorised as 1: first order, 2: second to fourth order: and 3: fifth order. Education was categorised as no education attained, primary level attained and the postprimary level attained; however, education level of at least primary was related to the reduced risks of perinatal and late neonatal mortality, so I later made it as a single dummy variable of no education (0: at least primary level attained and 1: no education level at all). Maternal age was grouped as <20 years, 20–29 years and 30 years and above based on the non-parametric analysis (figure 1 in online supplemental material 1).

### Data cleaning and organisation

Data analysis and cleaning such as categorisation and removing of duplicates were done in STATA V.15. An initial examination of the relationship between the key study independent variables and study outcomes was performed. Non-parametric generalised additive modelling on outcome variables and maternal age relationship was also done to guide the variable classification (figure 1 in online supplemental material 1), assuming a non-linear relationship between age and study outcomes.

Multicollinearity between the independent variables was considered at the cut-off point of Variance Inflation Factor (VIF) <10,[42] and there was no evidence of potential multicollinearity as each variable's VIF was less than 6 (table 1 online supplemental material 1). However, because of strong correction between the multiple birth and LBW, multiple birth was dropped under the perinatal mortality multivariable model, while LBW was dropped under the late neonatal mortality multivariable model. Because of the homogeneity across the study clusters, such as villages and subcounties within the HDSS, the clustering or multilevel modelling approach could not yield a variation across these clusters. Nonetheless, the use of a fixed-effect approach with the inclusion of variables such as place of residence (urban or rural) and household wealth index as key variables of interest provides an insight into the variation in newborn mortality across the place of residence and households.

### Multiple imputations for missingness and sensitivity analysis

The extent and pattern of missing data were scrutinised to guide the modelling strategy. Birth weight was missing among 27%, 19% for wealth index and <1% for other variables. To check if the missingness status was missing completely at random or missing at random, I ran a model of missing dummies (0: not missing and 1: missing), controlling for variables that I expected to increase the probability of missingness. For birth weight, I found that the likelihood of missing was highly associated with those who did not deliver in the health facilities and those who were born as stillbirths or died immediately after birth (table 2 in online supplemental material 1), indicating a possibility of birth weight missing at random. Similarly, the likelihood of wealth index was highly associated with the maternal age of less than 20 years (table 2 in online supplemental material 1), indicating a possibility of wealth index missing at random. Given the determinants of missingness, I ran multiple imputations (m=100), controlling for identified factors that contribute to the likelihood of missingness and all other variables (as auxiliary) in the dataset. The multiple imputation method has been indicated as an important approach for minimising the missingness bias,[43] and this does not depend on the magnitude of missingness.[44 45] After imputation, LBW was found to be underestimated for completed cases data, while the household's poor status was found to be overestimated for completed cases data (figure 2 in online supplemental material 1).

Additionally, considering newborn mortality as a rare event, I ran a model for rare events known as the firth logistic model[46] to assess the sample size bias for perinatal mortality, late neonatal mortality and LBW coefficient estimates. Comparing the firth logit model results with the logistic regression model, I found that the results were consistent (table 3 in online supplemental material 1). I later decided to use logistic modelling because of the less time it takes while running imputed data than the firth logit.

### Statistical modelling

The analysis was guided by this study's newborn framework (figure 3 in online supplemental material 1) that was generated based on the available variables in the dataset and literature on the factors associated with newborn mortality and LBW. The outcome variables (LBW, prenatal mortality and late neonatal mortality) were in the form of binary, with '1' indicating the presence of outcome exposure and '0' indicating the absence of outcome exposure. I, therefore, used a generalised linear modelling approach with a logit link function. To examine the newborn mortality pathway, several analysis steps were applied. Bivariate model(s) with all covariates on the perinatal, late mortality and LBW as outcome variables were performed to identify the variables that would significantly affect outcomes of interest in the

adjusted models. The results were reported in terms of ORs and 95% CI. The level of significance was determined at a p value of ≤0.05. Details on the analysis of the mediation effect of LBW on newborn mortality are indicated in online supplemental material 2.

### Patient and public involvement

There was no patient involvement in the study.

### RESULTS

#### Participants' characteristics

Of the total registered births between 2011 and 2015, 64% were rural residents (online supplemental material 3). The average maternal age was 27.3 years (SD=±6.5), and 14% were adolescent mothers. Overall, 84% of deliveries occurred in health facilities, of which 15% occurred in private health facilities (online supplemental material 3). Regarding maternal socioeconomic, 86% of the mothers had attained at least primary level education, 53% belonged to a wealth index of 3–5 and 86% were staying with a partner or married (online supplemental material 3).

#### Perinatal mortality, neonatal mortality and LBW estimates

The overall perinatal mortality rate and stillbirth rate within 7 days after delivery (0–6) in the 5 years preceding 2016 were 31 and 12 per 1000 total births, respectively (online supplemental material 3). Of the total prenatal mortality cases, death within 24 hours (0–1 days+stillbirths) accounted for 83%, while death within 1–6 days accounted for 30.8% (figure 1). Similarly, the overall neonatal mortality rate in the 5 years preceding 2016 was 22 per 1000 live births. Of the neonatal mortality cases, 62% were deaths within 24 hours of life (0–1 day), and 14 per 1000 live births were deaths within 7–27 days (figure 1).

Comparing the neonatal and perinatal mortality across 5 years of birth and death outcome registration, we see a slight change in the perinatal mortality curve between 2011 and 2014, with the mortality rate reducing from 31 to 28 per 1000 total birth and gaining its position in the subsequent years (figure 2). The newborn mortality within day 0–1 has also not significantly changed (27 per 1000 in 2011 vs 26 per 1000 in 2015). Similarly, slight changes in neonatal mortality rates were observed between 2013 and 2015, with the rates increasing by three units between 2013 and 2014 and reducing by six units in the subsequent year (figure 2). The newborn mortality within day 1–6 has also not significantly changed (10 per 1000 in 2011 vs 11 per 1000 in 2015). Regarding the LBW, we observe a linear trend between 2011 and 2012 that was later reduced by three units in the subsequent year – remaining uniform for the rest of the reporting years (figure 2).

#### Perinatal mortality, neonatal mortality and LBW distribution by social-economic and demographic characteristics

I examined the study's LBW distribution, perinatal and neonatal mortality by socioeconomic and demographic

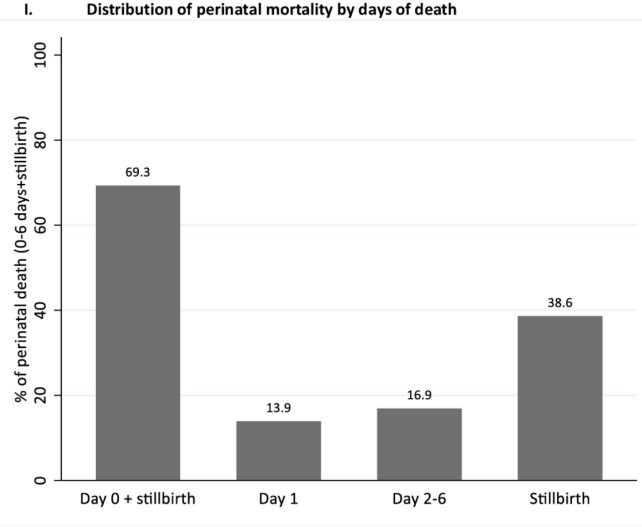

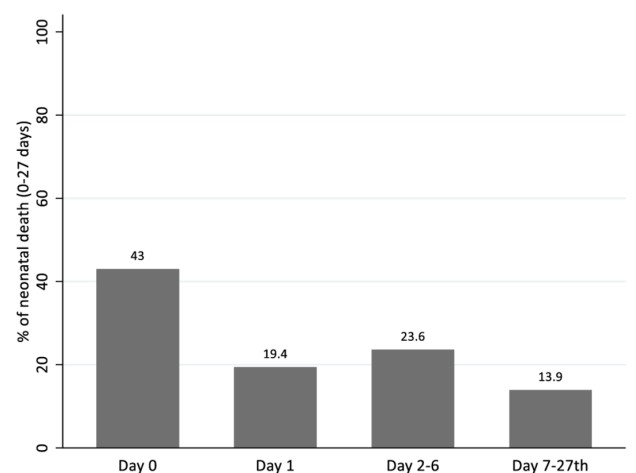

**Figure 1** Perinatal and newborn mortality distribution by days of death.

characteristics. LBW was higher among adolescent aged women than those aged 20 years and above, unmarried women and rural area residents (figure 3). Perinatal mortality was higher among those who delivered in the community and public health facilities. For neonatal mortality, a slightly lower death among married, urban residents and deliveries in the private health facilities was observed (figure 3). Additionally, LBW and neonatal mortality were observed to be higher among those whose mothers had ever experienced pregnancy loss and those born as multiple births (figure 4). The previous experience of pregnancy loss was also related to increased cases of perinatal mortality (figure 4). The potential factors for perinatal mortality, late neonatal mortality and LBW are investigated in the subsequent sections.

#### The perinatal mortality pathways

To describe the perinatal mortality pathway, I provide the results in two tables (tables 1 and 2). Table 1 indicates how the association of LBW with perinatal mortality changes with institutional delivery and the inclusion of other factors. Table 2 shows the indirect

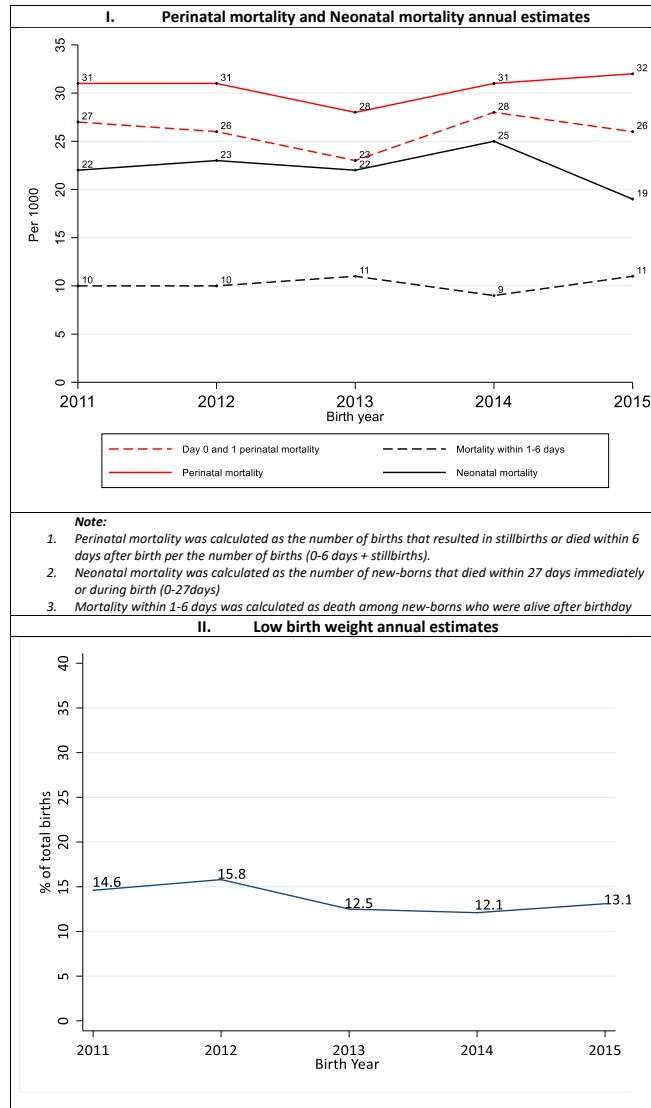

**Figure 2** Annual changes in perinatal mortality rates, neonatal mortality rates and LBW. LBW, low birth weight.

factors associated with perinatal mortality via LBW. At the bivariable level, the crude odds of perinatal mortality were slightly higher for facility deliveries relative to those conducted in the community (OR=1.01, 95% CI 0.80 to 1.28) (table 1). Also, at the bivariable level, the crude odds of perinatal mortality were higher among LBW babies relative to those weighing 2.5 kg+ (OR=2.54, 95% CI 1.74 to 3.70). Controlling for health facility delivery and introducing an interaction between LBW, a modest effect of health facility delivery on the effect of LBW was observed (OR=0.91, 95% CI 0.40 to 2.07) (table 1). Controlling for other factors in the final model, the odds of perinatal mortality among the LBW newborns remained almost the same (OR=2.55, 95% CI 1.15 to 5.67). The other factors that were directly associated with the increased odds of perinatal mortality were being a rural resident (OR=1.38, 95% CI 1.02 to 1.85), mother's previous experience of perinatal mortality (OR=3.95, 95% CI 2.86 to 5.46), advanced maternal age

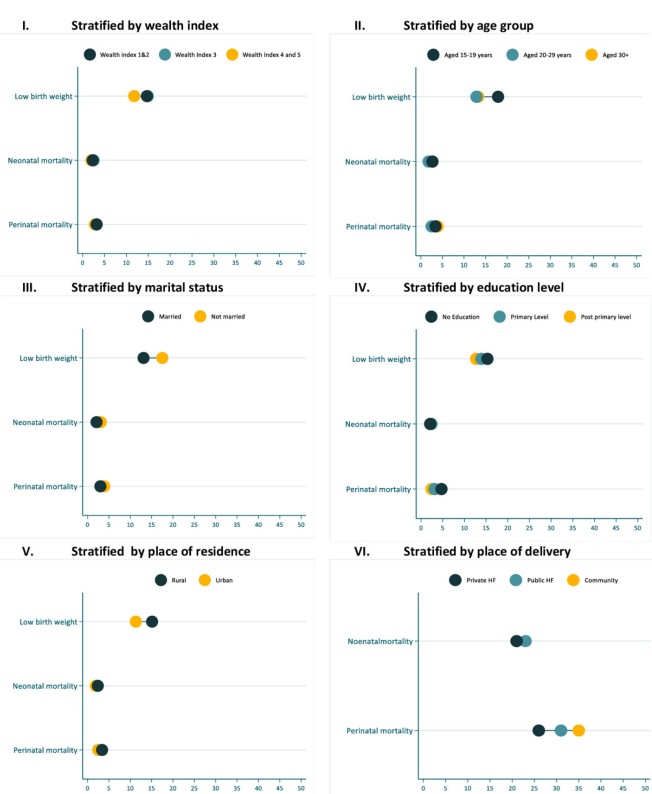

**Figure 3** Newborn mortality and LBW stratified by maternal socioeconomic characteristics, place of residence and place of delivery. LBW, low birth weight.

(OR=1.68, 95% CI 1.21 to 2.33) and having no education at all (OR=1.63, 95% CI 1.21 to 2.18). Birth of fifth order and above was inversely associated with perinatal mortality (OR=0.38, 95% CI 0.17 to 0.87) (table 1).

As mentioned earlier, LBW was strongly associated with perinatal mortality and was considered a potential mediating factor for perinatal mortality. Table 2 indicates the magnitude effect of the LBW factors in the study's dataset. The factors that were associated with

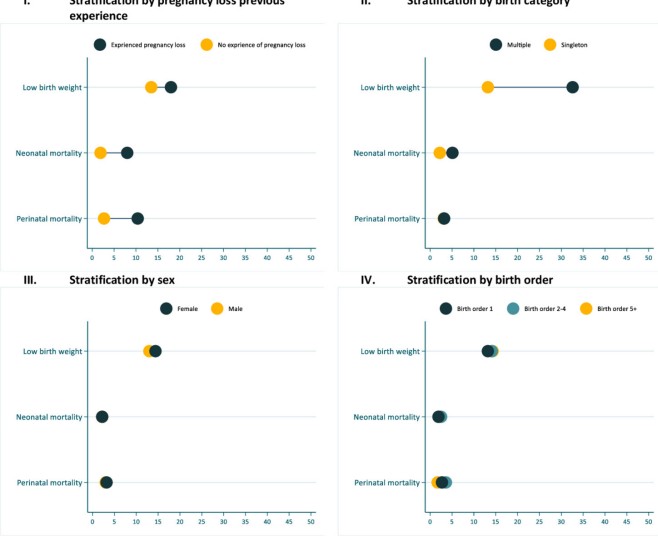

**Figure 4** Newborn mortality and LBW stratified by individual characteristics. LBW, low birth weight.

**Table 1** Underlying determinants of perinatal mortality using 2011–2015 Iganga-Mayuge HDSS event histories data

| | Unadjusted | | Adjusted 1 | | Adjusted 2 | | Fully adjusted | |
|---|---|---|---|---|---|---|---|---|
| | OR | 95% CI | OR | 95% CI | OR | 95% CI | OR | 95% CI |
| Childbirth weight | | | | | | | | |
| <2.5 kg | 1.00 | – | 1.00 | – | 1.00 | – | 1.00 | – |
| 25 kg+ | 2.54 | (1.74 to 3.70)*** | 2.54 | (1.75 to 3.71)*** | 2.73 | (1.21 to 6.14)* | 2.55 | (1.15 to 5.67)* |
| Place of delivery | | | | | | | | |
| Community | 1.00 | – | 1.00 | – | 1.00 | – | 1.00 | – |
| Health facility | 1.01 | (0.8 to 1.28) | 1.04 | (0.82 to 1.32) | 1.06 | (0.82 to 1.37) | 1.16 | (0.89 to 1.53) |
| Place of delivery and LBW interaction** | | | | | | | | |
| <2.5 kg and health facility delivery interaction | – | – | – | – | 0.91 | (0.4 to 2.07) | 0.91 | (0.40 to 2.06) |
| Annual birth quarters | | | | | | | | |
| 1 | 1.00 | – | – | – | – | – | 1.00 | – |
| 2 | 1.08 | (0.78 to 1.49) | – | – | – | – | 1.07 | (0.77 to 1.49) |
| 3 | 1.26 | (0.93 to 1.72) | – | – | – | – | 1.23 | (0.9 to 1.68) |
| 4 | 1.20 | (0.87 to 1.64) | – | – | – | – | 1.20 | (0.87 to 1.66) |
| Maternal education level | | | | | | | | |
| Primary+ | 1.00 | – | – | – | – | – | 1.00 | – |
| None | 1.69 | (1.27 to 2.25)*** | – | – | – | – | 1.63 | (1.21 to 2.18)*** |
| Maternal age (year) | | | | | | | | |
| <20 | 1.12 | (0.83 to 1.53) | – | – | – | – | 1.08 | (0.50 to 2.33) |
| 20–29 | 1.00 | – | – | – | – | – | 1.00 | – |
| 30+ | 1.48 | (1.18 to 1.85)*** | – | – | – | – | 1.68 | (1.21 to 2.33)*** |
| Childbirth order | | | | | | | | |
| 1 | 0.79 | (0.62 to 1.01) | – | – | – | – | 1.05 | (0.69 to 1.60) |
| 2–4 | 1.00 | – | – | – | – | – | 1.00 | – |
| 5th+ | 0.51 | (0.23 to 1.14) | – | – | – | – | 0.38 | (0.17 to 0.87)* |
| Place of residence | | | | | | | | |
| Urban | 1.00 | – | – | – | – | – | 1.00 | – |
| Rural | 1.40 | (1.1 to 1.78)** | – | – | – | – | 1.38 | (1.02 to 1.85)* |
| Marital status | | | | | | | | |
| No partner | 1.00 | – | – | – | – | – | 1.00 | – |
| Has a partner | 0.74 | (0.54 to 1.02) | – | – | – | – | 0.75 | (0.52 to 1.07) |

Continued

**Table 1** Continued

| | Unadjusted | | Adjusted 1 | | Adjusted 2 | | Fully adjusted | |
|---|---|---|---|---|---|---|---|---|
| | OR | 95% CI | OR | 95% CI | OR | 95% CI | OR | 95% CI |
| Experienced neonatal or pregnancy loss previously | | | | | | | | |
| No | 1.00 | – | – | – | – | – | 1.00 | – |
| Yes | 4.18 | (3.1 to 5.63)*** | – | – | – | – | 3.95 | (2.86 to 5.46)*** |
| Household wealth | | | | | | | | |
| Poor(er) (1–2) | 1.00 | | – | – | – | – | 1.00 | – |
| Less poor (3–5) | 1.10 | (0.88 to 1.38) | – | – | – | – | 0.85 | (0.65 to 1.12) |
| Child sex | | | | | | | | |
| Male | 1.00 | – | – | – | – | – | 1.00 | – |
| Female | 1.07 | (0.82 to 1.40) | – | – | – | – | 1.05 | (0.8 to 1.37) |
| First birth order–adolescence age interaction† | | | | | | | | |
| First order and <20 years | – | – | – | – | – | – | 1.18 | (0.49 to 2.87) |
| First birth order–advanced age interaction† | | | | | | | | |
| First order and 30+ years | – | – | – | – | – | – | 0.74 | (0.36 to 1.50) |
| Birth category | | | | | | | | |
| Singleton | 1.00 | – | – | – | – | – | – | – |
| Multiple | 1.02 | (0.5 to 2.08) | – | – | – | – | – | – |

*P<0.05; **p<0.01; ***p<0.001.
†Base – community delivery – birth weight 2.5 kg and above interaction.
†Base – second and above birth order – 30 years of age and below interaction.
‡Base – second and above birth order – 30 years of age and below interaction.
HDSS, Health Demographic and Surveillance Site; LBW, low birth weight.

**Table 2** Perinatal mortality indirect factors mediated through LBW using Iganga-Muyuge HDSS event history data

| | Unadjusted | | Adjusted | |
|---|---|---|---|---|
| | OR | 95% CI | OR | 95% CI |
| Annual birth quarters | | | | |
| 1 | 1.00 | – | 1.00 | – |
| 2 | 1.11 | (0.93 to 1.34) | 1.12 | (0.93 to 1.34) |
| 3 | 1.05 | (0.87 to 1.27) | 1.06 | (0.88 to 1.28) |
| 4 | 1.03 | (0.86 to 1.24) | 1.03 | (0.85 to 1.24) |
| Maternal education level | | | | |
| Primary+ | 1.00 | – | 1.00 | – |
| None | 1.14 | (0.95 to 1.37) | 1.14 | (0.94 to 1.37) |
| Maternal age (year) | | | | |
| <20 | 1.43 | (1.2 to 1.7)*** | 1.40 | (1.14 to 1.71)*** |
| 20−29 | 1.00 | – | 1.00 | – |
| 30+ | 0.94 | (0.82 to 1.08) | 0.98 | (0.84 to 1.14) |
| Childbirth order | | | | |
| 1 | 0.91 | (0.8 to 1.04) | 0.88 | (0.75 to 1.02) |
| 2−4 | 1.00 | – | 1.00 | – |
| 5th+ | 1.03 | (0.75 to 1.41) | 0.97 | (0.69 to 1.35) |
| Place of residence | | | | |
| Urban | 1.00 | – | 1.00 | – |
| Rural | 1.39 | (1.22 to 1.59)*** | 1.40 | (1.19 to 1.65)*** |
| Marital status | | | | |
| No partner | 1.00 | – | 1.00 | – |
| Has a partner | 0.71 | (0.6 to 0.85)*** | 0.76 | (0.63 to 0.92)** |
| Experienced neonatal or pregnancy loss previously | | | | |
| No | 1.00 | – | 1.00 | – |
| Yes | 1.45 | (1.11 to 1.89)** | 1.33 | (1.01 to 1.76)* |
| Birth category | | | | |
| Singleton | 1.00 | – | 1.00 | – |
| Multiple | 3.10 | (2.31 to 4.16)*** | 3.12 | (2.31 to 4.21)*** |
| Household wealth | | | | |
| Poor(er) (1–2) | 1.00 | – | 1.00 | – |
| Less poor (3–5) | 1.16 | (1.02 to 1.32)* | 0.92 | (0.78 to 1.08) |
| Child sex | | | | |
| Male | 1.00 | – | 1.00 | – |
| Female | 1.13 | (0.99 to 1.29) | 1.12 | (0.99 to 1.28) |

*P<0.05; **p<0.01; ***p<0.001.
HDSS, Health Demographic and Surveillance Site.

the increased odds of LBW were adolescent mothers (OR=1.40, 95% CI 1.14 to 1.71), rural resident mothers (OR=1.40, 95% CI 1.19 to 1.65), mothers with previous experience of newborn or pregnancy loss (OR=1.33, 95% CI 1.01 to 1.76) and babies born as multiple births (OR=3.12, 95% CI 2.31 to 4.21). Having a partner or being married was inversely associated with LBW (OR=0.76, 95% CI 0.63 to 0.92) (table 2). Based on tables 1 and 2 results, the indirect effects are calculated as:

$$indirect\ effect_i = \beta_{lbwi} * \beta_{lbw}\ x_{lbw} \quad (1)$$

Total effects are calculated as:

$$total\ effect = \frac{\beta_{lbwi} * \beta_{lbw}\ x_{lbw}}{\beta_{lbwi} * \beta_{lbw}\ x_{lbw} + \beta_{pmi}} \quad (2)$$

where, $\beta_{lbw}\ x_{lbw}$ represents LBW coefficient in PM model, $\beta_{lbwi}$ represents variable coefficient in the LBW model and $\beta_{pmi}$ represents variable coefficient in the PM model that matches $\beta_{lbwi}$. Table 1 presents results on direct factors of PM and table 2 presents results on indirect factors of PM that are mediated through LBW.

Based on the two equations, LBW mediated +47% of rural resident mothers, +54% of adolescent mothers,

**Table 3** Late neonatal mortality risk factors using 2011–2015 Iganga-Mayuge HDSS event histories data

| | Unadjusted | | Adjusted | |
|---|---|---|---|---|
| | OR | 95% CI | OR | 95% CI |
| Annual birth quarters | | | | |
| 1 | 1.00 | – | 1.00 | – |
| 2 | 1.55 | (0.55 to 2.91) | 1.65 | (0.58 to 4.68) |
| 3 | 1.63 | (0.59 to 3.02) | 1.64 | (0.59 to 4.56) |
| 4 | 1.43 | (0.49 to 2.72) | 1.41 | (0.49 to 4.09) |
| Education level | | | | |
| No level of education | 1.00 | – | 1.00 | – |
| Primary education+ | 1.37 | (0.56 to 2.35) | 1.24 | (0.51 to 3.06) |
| Maternal age (years) | | | | |
| <20 | 0.87 | (0.31 to 1.66) | 1.21 | (0.37 to 3.97) |
| 20–29 | 1.00 | – | 1.00 | – |
| 30+ | 1.73 | (0.87 to 2.63) | 1.73 | (0.81 to 3.69) |
| Childbirth order | | | | |
| 1st | 0.70 | (0.34 to 1.07) | 0.88 | (0.38 to 2.04) |
| 2−4 | 1.00 | – | 1.00 | – |
| 5th+ | 2.20 | (0.67 to 4.54) | 1.89 | (0.54 to 6.61) |
| Place of residence | | | | |
| Urban | 1.00 | – | 1.00 | – |
| Rural | 0.77 | (0.38 to 1.17) | 0.82 | (0.35 to 1.95) |
| Marital status | | | | |
| Has no partner | 1.00 | – | 1.00 | – |
| Has a partner | 0.66 | (0.27 to 1.14) | 0.54 | (0.21 to 1.43) |
| Experienced neonatal or pregnancy loss previously | | | | |
| No | 1.00 | – | 1.00 | – |
| Yes | 3.60 | (1.39 to 6.46)** | 3.17 | (1.15 to 8.74)* |
| Birth category | | | | |
| Singleton | 1.00 | – | 1.00 | – |
| Multiple | 7.71 | (2.95 to 13.85)*** | 6.93 | (2.58 to 18.55)*** |
| Household wealth | | | | |
| Poorer (1–2 index) | 1.00 | – | 1.00 | – |
| Less poor (3–5 index) | 0.75 | (0.37 to 1.16) | 0.72 | (0.29 to 1.74) |
| Child sex | | | | |
| Male | 1.00 | – | 1.00 | – |
| Female | 0.57 | (0.28 to 0.88) | 0.55 | (0.27 to 1.12) |
| Place of delivery | | | | |
| Community | 1.00 | – | – | – |
| Health facility | 1.22 | (0.57 to 1.95) | – | – |

*p<0.05; **p<0.001; ***p<0.001.
HDSS, Health Demographic and Surveillance Site.

+15% of mothers with previous experience of new-born or pregnancy loss, +100% of multiple births and -45% of mothers with partners.

The increase in the odds of late neonatal mortality was associated with previous experience of pregnancy loss or neonatal mortality (OR=3.17, 95% CI 1.15 to 8.74) and multiple births (OR=6.93, 95% CI 2.58 to 18.55) (table 3).

## DISCUSSION

To the best of my knowledge, this is the first study in a resource-poor setting that seeks to understand how LBW newborn mediates newborn mortality and how the institution delivery may moderate the LBW effect on newborn survival. The analysis was based on an event history health demographic dataset that has not been exploited in studying newborn survival in Uganda and

other countries with the same context. The study results provide an insight into newborn survival life course and health systems interventions, particularly targeting the prevention of LBW occurrences and management of or care for LBW newborns. In the subsequent subsections, I discuss: (1) the health facility pathways in affecting the survival of newborns and (2) the pathway through which LBW mediate the newborn survival. The results are interpreted, and where possible, recommendations for the design of appropriate interventions are discussed based on the available studies and implementation research evidence in a similar context.

### Health facility delivery and neonatal mortality

The study results confirm the persistent high perinatal mortality rates in Uganda, with stillbirths and death within 24 hours contributing the largest share. The modest effect of health facility delivery on the survival of LBW newborns and the largest proportion of newborn death within 24 hours after birth indicates a gap in accessing the required health services. We know that, during labour, women should be attended to by skilled health workers who are knowledgeable and qualified to screen for labour complications such as obstructed labour conditions, which need life-saving emergency services, for instance, caesarean section.[32 47] However, access to these interventions is usually limited in such resource-limited setting.[3] Furthermore, the substantial effect of LBW on the survival of newborns is well documented,[10 12] but interventions for preventing or controlling the effect of LBW are always inadequately available in such resource-limited settings. For instance, kangaroo mother care and resuscitation that are easy to implement, inexpensive and effective[1 47] for helping such newborns survival are not universally implemented in most of the health facilities in such settings. Additionally, access to antenatal care such as screening for maternal morbidities and prenatal interventions may help protect the mother–fetus dyad from maternal morbidities[48] and ultimately reduce the likelihood of adverse outcomes. However, usually, most women do not access such services.[37]

Noteworthy is that in this study's setting and SSA in general, such interventions may fail to work effectively because of community contextual problem, in particular poor transport and referral systems, which contribute to delays and failures in receiving required services. For instance, there is one public hospital (Iganga Hospital) in this study setting that provides comprehensive emergency obstetric and newborn care services in the district.[49] Because of the long-distance coupled with poor transport and referral systems, women from rural or remote areas usually fail to access the services and those that make it reach when they are in advanced stages of complications. Such could perhaps explain why rural residents were associated with a high risk of perinatal mortality in this study.

Furthermore, in such settings, most women first access services at different care points within the community because of pluralism in healthcare. They are later referred or go to the accredited or higher level facilities when adverse complications have emerged. Thus, most women who deliver in higher level facilities often have complications associated with a high likelihood of perinatal mortality. These highlighted challenges could explain why health facility delivery could not contribute substantially to the reduced probability of stillbirth and death within 24 hours. Additionally, the rural communities are characterised by cultural and social behaviours that affect access to pregnancy, delivery and newborn care interventions. These findings suggest interventions that simultaneously improve healthcare access coverage and quality of care if the facility interventions match better outcomes.

### LBW effect on perinatal and neonatal mortality

LBW was identified as a potential determinant of perinatal mortality. The other determinants of newborn mortality were marital status, maternal age and maternal education. Having a partner was inversely associated with LBW, which was consistent with another study on the effect of marital status on the birth outcome,[50] implying that it may reduce perinatal mortality indirectly through birth weight. The effect of marital status on child health outcome could be via decision-making power, which could influence the choice of appropriate services. In the study's context and SSA, men usually possess resources and assets; thus, unmarried or single women access inadequate health services.[51] In addition to the negotiating power with the health workers, married women have support from their spouse in terms of joint income, joint care and joint decision making, which may contribute to their birth preparedness,[52] child nutrition and general health of the household.

Furthermore, the health policy or rule in Uganda that require women to attend maternal services in accompany of their spouses may limit the unmarried counterpart from accessing maternal health facility services[53–55] that would address LBW. Studies done in similar settings have indicated that women would be denied services unless they attended with their husbands. Sometimes, those with husbands would be attended to as a priority.[53–55] While male engagement is a positive factor, it has been found that it could end up being a discriminative strategy that could create stigma among unmarried women, inhibiting them from accessing the required services.[54 55] Inclusive interventions that target all women groups regardless of statuses, such as age and marital status, should be designed.

Maternal age of 30 years+ was positively associated with perinatal mortality as the adolescence age (15–19 years) was positively associated with LBW, indicating how the tail ends of maternal age are associated with the increased likelihood of newborn mortality. Notwithstanding, the effect of maternal age on perinatal mortality needs to be interpreted with care because of other confounding factors such as birth order or parity. On the first hand, the effect of maternal age of 30 years+ could be related to birth order. Consistent with a study done in India and

Bangladesh,[56] newborns of birth order of at least five were at lower risk of mortality compared with the first birth order. When an interaction between maternal age and birth order was introduced, the first births among women aged at least 30 years were positively associated with perinatal mortality. However, we also know that high parity is associated with pregnancy complications such as hypertension in pregnancy, leading to an increased likelihood of adverse birth outcomes. In low-income and middle-income countries where maternal age at the first delivery begins as early as 10 years,[57] by the age of 25–30 years, most women have experienced at least four pregnancies that would increase the risk of hypertension and ultimately adverse birth outcomes. In this study, 28% and 49% of women aged 20–29 and 30 years respectively had children of third and fourth birth order. These results suggest adolescent pregnancy and fertility control interventions targeting adolescents, newly delivered women and multiparous women. Furthermore, the results suggest clinical screening and treatment for morbidity in pregnancy, which may reduce adverse birth outcomes.

In this study, multiple births were not strongly associated with perinatal mortality but strongly associated with mortality at a later age. Although some studies have indicated multiple births to be highly associated with infant and under-five mortality,[58 59] this study highlights that the association may vary by age. The association of multiple births with perinatal mortality in this study has been indicated to be mediated by LBW. The association of multiple births with LBW and preterm births has also been studied in other studies.[60 61] Women with multiple births need special care[62 63] in terms of nutrition to produce enough breast milk to feed the newborns.

Furthermore, multiple newborns are more likely to be very small newborns, and appropriate regular postnatal examinations are recommended.[60] However, for marginalised communities and families, such care could be inadequate because of limited resources. In addition, multiple birth children are in most cases preterm or very LBW newborn and thus susceptible to recurrent infections that may need frequent healthcare services, which are often inadequately available in rural communities. Interventions within community and health facility levels for supporting women with multiple births are thus needed. I also recommend a comprehensive study on the care for multiple births, including small newborns, beyond the facility interventions, particularly focusing on the burden and the community practices of caring for such newborns in a resource-limited setting.

Attending at least primary education level was directly associated with the reduced likelihood of perinatal mortality, a finding that has been indicated in another study.[64] The pathway could be better health behavioural practices such as nutrition, better sanitation and appropriate healthcare access. In most cases, educated women have access to the health information on maternal nutrition, good hygiene practices and maternal danger signs that could have been attained either while at school or through reading media publications. These findings suggest a need to design and implement behavioural interventions that integrate less educated communities. One of the strategies that have been effective in mobilising and sensitising poor and illiterate communities in such resource-limited settings is the use of community health workers, which could be exploited to target such groups with maternal and newborn-specific health information. Additionally, interventions that promote girl child education and those that prevent early pregnancies targeting girls and families could increase education level among women in the long run.

Previous stillbirth and neonatal death experience was related to a high likelihood of perinatal mortality and continued to affect the neonates in the later stage, which is consistent with other studies done in developed countries.[65 66] Women with previous experience of pregnancy loss are susceptible to risk conditions such as pre-eclampsia, preterm births, intrauterine growth retardation and fetal distress that could affect the health of subsequent pregnancies.[67] Additionally, perinatal mortality recurrence, particularly stillbirths, could be related to chronic maternal conditions.[66] Proper screening for the histories of pregnancy loss and neonatal mortality in addition to the histories of related risk factors such as hypertension, diabetes and epilepsy should be emphasised during the prenatal and labour period.

### Study strengths and limitations
The study used a large event history health demographic data that had never been exploited to study perinatal and newborn health. The health and demographic surveillance data have been indicated to have a high statistical power close to 100% for mortality measurement.[68] Nevertheless, this study has some limitations related to the data source and context. First, gestation age data that could show if the birth was premature or otherwise and if the pregnancy loss/death was a stillbirth or otherwise were not documented during birth registration. Second, there was no information on the maternal morbidities and complications that lead to different birth outcomes, which could have been used to understand their effect along the path. Third, there was a considerable proportion of missingness for some of the variables that could lead to bias; however, multiple imputations approach was performed to minimise the missingness bias. Lastly, the study results are generalisable to eastern Uganda's setting. Nonetheless, since most communities in Uganda and SSA share the same community and health characteristics, these results could guide the design of health systems interventions that address mortality causes and risk factors.

### CONCLUSION
This manuscript contributes to understanding child mortality by analysing the newborn mortality pathways in a resource-poor setting. The study affirms the role of LBW in mediating the association of socioeconomic and

demographic factors with newborn mortality. The study has also identified the continued effect of multiple birth and the mother's previous pregnancy loss experiences on newborns survival at a later age. Furthermore, in addition to the large proportion of mortality that occurs within the first day (24 hours), the study has discovered that institutional deliveries had a modest moderation effect in the association of LBW with newborn mortality.

These results also demonstrate the need for a life course approach in the design of interventions. These interventions should include control of LBW and management or treatment of LBW newborns. The LBW control interventions should target the socially and economically disadvantaged women, who include adolescents, unmarried women and rural residents, with information on birth preparedness and care for pregnancies. I propose a postnatal treatment and care package for mothers and health workers throughout the newborn life course for the LBW newborns and multiple birth. Intensifying the health systems by replicating known LBW effective strategies such as identifying LBW cases through immediate newborn weighing and their management through kangaroo mother care, better nutrition and other medical interventions are needed.

Additionally, the effect of institutional delivery in such resources limited setting is complex to understand because of the plural health services' points that women go through before reaching the formal care points. The challenges of inadequate amenities for maternal and newborn health, including qualified staff, may make it difficult for the health facilities to address obstetric emergencies appropriately. I suggest a comprehensive health systems' audit, both internal and external, to provide an insight into the reasons for the insignificant effect of institutional deliveries on the survival of LBW newborns and the large proportion of death within the first day of life.

**Acknowledgements** Drs Tiziana Leone and Arjan Gjonca advised on the study's design and frequently reviewed the statistical approaches in addition to reviewing the manuscript. Miss Tryphena Nareeba, Dr Dan Kajungu and Dr Peter Waiswa supported me to access the Iganga-Mayuge Health Demographic and Surveillance Site (HDSS) data. Dr Peter Waiswa also gave me valuable comments that improved this manuscript. I am also grateful for the HDSS data collection team and the mothers who tirelessly provide such information.

**Contributors** As a sole author of this manuscript, I solely did the analysis and wrote the manuscript.

**Funding** This study is part of my PhD project that is being funded by the London School of Economics and Political Science.

**Disclaimer** The funder had no role in study design, data collection and analysis, decision to publish or manuscript preparation.

**Competing interests** None declared.

**Patient consent for publication** Not required.

**Ethics approval** This study used anonymised secondary data that were provided and thus the patient consent for publication was not required.

**Provenance and peer review** Not commissioned; externally peer reviewed.

**Data availability statement** Data may be obtained from a third party and are not publicly available. The Iganga-Mayuge HDSS data are not shared publicly but can be made available to other researchers with approval by the Iganga-Mayuge

HDSS research ethics committee. Requests for data access may be sent to Collins Gyezayo at info@muchap.mak.ac.ug.

**ORCID iD**
Rornald Muhumuza Kananura http://orcid.org/0000-0002-9915-1989

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
