## [Reviewer comments · BMJ Open]

ARTICLE DETAILS

TITLE (PROVISIONAL)	The mediation role of low birth weight on the factors associated with new-born mortality and the moderation role of institutional delivery in the association of low birth weight with new-born mortality in a resource-poor setting
AUTHORS	Kananura, Rornald

VERSION 1 – REVIEW

REVIEWER	Rajat Das Gupta University of South Carolina
REVIEW RETURNED	29-Nov-2020

GENERAL COMMENTS	Kananura aimed to assess the mediation effect of Low Birth Weight (LBW) on New-born Mortality (NM) and the role of institutional delivery in their association. Using the 2011-2015 event histories health demographic data collected by Iganga-Mayuge Health Demographic and Surveillance Site (HDSS), the author find that LBW was positively associated with perinatal mortality. The factors that were indirectly associated with perinatal mortality via LBW were adolescence age, rural residence, mothers' previous experience of NM, multiple births, and marital status. Multiple births and mother's prior experience of NM were positively associated with NM at a later age. Institutional delivery had a modest inverse role in the association of LBW with NM. The article has potential merit. However, I have several comments/concerns: 1. In the introduction section, the author mentioned "However, the newborn mortality rate shares the largest portion of under-five mortality and has remained unacceptably high in Sub-Saharan Africa (SSA)." It would be worth to mention the percentage of the under-five mortality attributed to newborn mortality.2. Before the methods section, the authors should present a Directed acyclic graph/ DAG to visualize the variables including the mediators.3. The author merged unmarried and divorced group as single category. Are these two groups similar in characteristics?4. Please mention the level of significance in the statistical analysis section.5. Please mention what measures of association was used (i.e.: Prevalence Ratio/ Odds Ratio)?6. In the result section the authors mentioned, "Similarly, the overall neonatal mortality rate in the 5 years preceding 2016 was 22 per 1000 live births. Of the neonatal mortality cases, 62% were deaths within 24 hours of life (0-1 day) and 14 per 1000 live births were deaths within 7-27 days (Figure 1)." My comment is it will
--

	also be interesting to learn the deaths per 1000 live births within 0-1 days and 1-7 days. 7. In the result section the authors wrote, “Comparing the neonatal and perinatal mortality across five years of birth and outcome registration, we see a slight change in perinatal mortality curve between 2012-2014 with mortality rate reducing from 31-28 per 1000 total birth and gaining its position in the subsequent years (Figure 2).” Please change “from 31 to 28 per 1000 total birth.” 8. In the tables please report Odds Ratio with 95% Confidence Interval instead of co-efficient. 9. Please mention the Ethical considerations of the study.
--	---

REVIEWER	Anu Thukral All India Institute of Medical Sciences, New Delhi, India
REVIEW RETURNED	18-Dec-2020

GENERAL COMMENTS	We read with interest the association of low birth weight with new-born mortality in a resource-poor setting: a pathway analysis of event history demographic data by Rornald Muhumuza Kananura. Abstract The Iganga-Mayuge HDSS data is located eastern Uganda. The author probably means The Iganga-Mayuge HDSS data is located in eastern Uganda. Outcome: The study’s key outcomes or endogenous factors that were; the author probably means The study’s key outcomes or endogenous factors were. Outcome Measure: One also needs to specify that additional objectives were to evaluate the factors associated with LBW, PM, NM and so on in the abstract as well. Introduction The author mentions that there is dearth of literature on mediation of LBW and mortality; the following articles however give an insight into the same and should be referenced.  1. Association of Low Birth Weight Infants and Maternal Sociodemographic Status in Tuzla Canton during 1992–1995 War Period in Bosnia and Herzegovina 2. Neonatal mortality of low-birth-weight infants in Bangladesh. S Yasmin, D Osrin, E Paul, A Costello Ethics Missing
--

VERSION 1 – AUTHOR RESPONSE

Reviewer: 1

Dr. Rajat Das Gupta, BRAC University James P Grant School of Public Health Comments to the Author:

Kananura aimed to assess the mediation effect of Low Birth Weight (LBW) on New-born Mortality (NM) and the role of institutional delivery in their association. Using the 2011-2015 event histories health demographic data collected by Iganga-Mayuge Health Demographic and Surveillance Site (HDSS), the author find that LBW was positively associated with perinatal mortality. The factors that were indirectly associated with perinatal mortality via LBW were adolescence age, rural residence, mothers’ previous experience of NM, multiple births, and marital status. Multiple births and mother’s prior experience of NM were positively associated with NM at a later age. Institutional delivery had a modest inverse role in the association of LBW with NM. The article has potential merit. However, I have several comments/concerns:

Comment 1: In the introduction section, the author mentioned “However, the newborn mortality rate shares the largest portion of under-five mortality and has remained unacceptably high in Sub-Saharan

Africa (SSA).” It would be worth to mention the percentage of the under-five mortality attributed to newborn mortality.

Response: I have included the percentages based on the recent UN 2020 estimates. The sentence now reads as “However, the new-born mortality continues to share the largest portion of under-five mortality (47%) and has remained unacceptably high in Sub-Saharan Africa (SSA) [4–7]” – Page 1, first paragraph

Comment 2: Before the methods section, the authors should present a Directed acyclic graph/ DAG to visualize the variables including the mediators.

Response: While this is a good idea, I am limited by the journal’s accepted number of figures and tables. However, I have provided a graph describing the new-born mortality, with LBW as a mediating factor and place of delivery as a moderating factor in supplement 1 under includes a statement on how this was developed for guiding analysis under the statistical modelling. – Page 10, first paragraph

Comment 3: The author merged unmarried and divorced group as single category. Are these two groups similar in characteristics?

Response: I have changed this to having or not having a partner instead of marital status. The thinking is that those who currently have male partners may be supported in for of financial and social support and thus improving access to better health and wellbeing services. But also, the sample size for widowed and divorced was very small, and so combining them as having a partner (living together as if they are married) or not would be better.

Comment 4: Please mention the level of significance in the statistical analysis section.

Response: The level of significance has been provided as “The results were reported in terms of odds ratios (OR) and 95% confidence interval (CI), while the level of significance was determined at p-value of ≤ 0.05 ”. – Page 10.

Comment 4: Please mention what measures of association was used (i.e.: Prevalence Ratio/ Odds Ratio)?

Response: The level of association has also been provided as in comment 4.

Comment 5: In the result section the authors mentioned, “Similarly, the overall neonatal mortality rate in the 5 years preceding 2016 was 22 per 1000 live births. Of the neonatal mortality cases, 62% were deaths within 24 hours of life (0-1 day) and 14 per 1000 live births were deaths within 7-27 days (Figure 1).” My comment is it will also be interesting to learn the deaths per 1000 live births within 0-1 days and 1-7 days.

Response: I agree with the reviewer on indicating the 0-1 and 1-6 mortality per 1000. Figure 2 has been updated to include these categories.

Comment 6: In the result section the authors wrote, “Comparing the neonatal and perinatal mortality across five years of birth and outcome registration, we see a slight change in perinatal mortality curve between 2012-2014 with mortality rate reducing from 31-28 per 1000 total birth and gaining its position in the subsequent years (Figure 2).” Please change “from 31 to 28 per 1000 total birth.”

Response: I thank the reviewer for his suggestion. I have instead changed the period (2011-2014) since my interest is to show how the rates are changing.

Comment 7: In the tables please report Odds Ratio with 95% Confidence Interval instead of co-efficient.

Response: I have changed the regression results to Odds ratio and confidence intervals

Comment 8: Please mention the Ethical considerations of the study.

Response: A section on Ethics has been included as “This study used anonymised secondary data that were provided by Iganga-Mayuge Health Demographic and Surveillance site (HDSS) in central-eastern Uganda. The data were provided after formal approval by the HDSS research committee”.

Reviewer: 2

Dr. Anu Thukral, All India Institute of Medical Sciences Comments to the Author:

We read with interest the association of low birth weight with new-born mortality in a resource-poor setting: a pathway analysis of event history demographic data by Rornald Muhumuza Kananura.

Abstract

Comment 1: The Iganga-Mayuge HDSS data is located eastern Uganda. The author probably means The Iganga-Mayuge HDSS data is located in eastern Uganda. Outcome: The study's key outcomes or endogenous factors that were; the author probably means The study's key outcomes or endogenous factors were.

Response: I thank the reviewer for pointing this out. I have read and edited the manuscript for all typos and grammatical errors.

Comment 2: Outcome Measure: One also needs to specify that additional objectives were to evaluate the factors associated with LBW, PM, NM and so on in the abstract as well.

Response: Although the analysis looks at both factors associated with newborn death outcomes and low birth weight, my main interest is which factors are mediated by LBW to affect NM and PM. In fact, the analysis approach (pathway) considers all the factors (direct and indirect). Therefore, maintaining the study objective as "To assess the mediation effect of Low Birth Weight (LBW) on New-born Mortality (NM) including stillbirth and the role of institutional delivery in their association" caters for the reviewer's suggestions.

Introduction

Comment 3: The author mentions that there is dearth of literature on mediation of LBW and mortality; the following articles however give an insight into the same and should be referenced.

1. Association of Low Birth Weight Infants and Maternal Sociodemographic Status in Tuzla Canton during 1992–1995 War Period in Bosnia and Herzegovina
2. Neonatal mortality of low-birth-weight infants in Bangladesh. S Yasmin, D Osrin, E Paul, A Costello

Response: While a lot has been studied on the association of LBW with neonatal mortality, we still lack evidence on how LBW mediates some of the factors and how institutional delivery moderates the association of LBW and newborn mortality. In fact, most studies that have looked at the association between LBW and neonatal mortality have not considered how the LBW may affect the different newborn age groups since the impact of the magnitude of risk factors may shift with an increase in age groups so as to identify the critical period.

I have also read the references provided by the reviewer, and while they were important reads, they do not apply the analysis approach applied in this manuscript. For instance, a Study by Skokić et al. does not show how the effect of LBW on infancy may change by age bands.

Comment 4: Ethics Missing

Response: Thank you for pointing out this. I have now included a section on Ethical consideration.

VERSION 2 – REVIEW

REVIEWER	Das Gupta, Rajat BRAC University James P Grant School of Public Health
REVIEW RETURNED	12-Apr-2021
GENERAL COMMENTS	The authors have addressed all the comments. The quality of the manuscript increased. I recommend the manuscript for publication.